# The Potential Antidepressant Action of Duloxetine Co-Administered with the TAAR1 Receptor Agonist SEP-363856 in Mice

**DOI:** 10.3390/molecules27092755

**Published:** 2022-04-25

**Authors:** Xia Ren, Jiaying Xiong, Lingzhi Liang, Yin Chen, Guisen Zhang

**Affiliations:** 1School of Pharmacy, Jiangsu Ocean University, Lianyungang 222005, China; 2019220333@jou.edu.cn (X.R.); 2019220320@jou.edu.cn (L.L.); 2School of Medicine, Guangxi University of Science and Technology, Liuzhou 545005, China; jiayingxiong@gxust.edu.cn

**Keywords:** depression, duloxetine, SEP-363856, co-administration, antidepressant-like effect

## Abstract

Here, we explored the possible interaction between duloxetine and SEP-363856 (SEP-856) in depression-related reactions. The results showed that oral administration of duloxetine showed powerful antidepressant-like effects in both the forced swimming test (FST) and the suspension tail test (TST). SEP-856 orally administered alone also exerted an antidepressant-like effect in FST and TST, especially at doses of 0.3, 1, and 10 mg/kg. In addition, duloxetine (15 mg/kg) and SEP-856 (15 mg/kg) both showed antidepressant-like effects in the sucrose preference test (SPT). Most importantly, in the above experiments, compared with duloxetine alone, the simultaneous use of duloxetine and SEP-856 caused a more significant antidepressant-like effect. It is worth noting that doses of drug combination in FST and TST did not change the motor activities of mice in the open-field test (OFT). Thus, duloxetine and SEP-856 seem to play a synergistic role in regulating depression-related behaviors and might be beneficial for refractory depression.

## 1. Introduction

Major depressive disorder (MDD), which is one of the most important causes of dis-ability worldwide, has the characteristics of chronicity, recurrence, and high suicide rate [1,2]. In addition, 33% of patients develop resistance to antidepressant treatment, a condition known as treatment-resistant depression (TRD) [3]. At present, the pathogenesis of depression has been explained by many mechanisms; the monoaminergic disorder of the central nervous system has always been regarded as the main pathogenesis of depression. Monoamine neurotransmitter serotonin (5-HT), norepinephrine (NE), and dopamine (DA) have been the main adjustment substances of antidepressant drugs in the past few decades [4].

Duloxetine, one of the main clinical antidepressants, was approved by the Food and Drug Administration (FDA) to treat MDD as early as 2002. It is a 5-HT and NE dual reuptake inhibitor (SNRI). It can enhance the nerve conduction mediated by the 5-HT and NE transmitter system, and it has antidepressant and general anti-anxiety effects [5,6]. Clinically, the initial dose of duloxetine for the treatment of major depression is 60 mg/day, and the maximum dose may be increased to 120 mg/day. After 2–4 weeks of treatment, it can improve the symptoms of depression and anxiety in patients [7,8]. In preclinical experiments, duloxetine can reduce the immobility time in forced swimming experiments and tail suspension experiments, as well as improve the anhedonia in sugar water preference experiments [9,10,11]. However, many adverse reactions are related to duloxetine, especially in the case of high doses and long-term medication. For example, the main adverse reactions include negative effects on the cardiovascular system, central nervous system, endocrine system, and digestive system, as well as allergic reactions [12]. In addition, traditional antidepressants such as duloxetine have an effect latency between 1 and 4 weeks. About one-third of patients do not respond to their first antidepressant drug treatment. Therefore, drugs with stronger and more rapid antidepressant effects and fewer side-effects need to be found.

For the treatment of TRD, the main strategy is to switch to another antidepressant, to use two or more antidepressants, or to combine an antidepressant with an enhancer [13,14]. Studies have shown that antipsychotic monotherapy and adjuvant therapy both show better efficacy than placebo in the treatment of MDD. For example, clinical evidence suggests that the use of risperidone as an adjuvant therapy for TRD may improve the re-mission rate [15,16]; the average change in the total Montgomery–Åsberg Depression Rating Scale (MADRS) score of 300 mg/day extended-release quetiapine combined with antidepressants compared with placebo combined with antidepressants is statistically significant. Aripiprazole (2–20 mg/day) combined with selective serotonin reuptake inhibitor (SSRI)/serotonin and norepinephrine reuptake inhibitor (SNRI) treatment of MADRS total score from baseline to the endpoint of the average change was statistically greater, and the mean shift was statistically significant. A study of co-administration of olanzapine and fluoxetine showed a statistically significant mean shift from baseline to endpoint in total MADRS scores compared with the use of fluoxetine alone [17].

Trace amine-related receptor 1 (TAAR1) is a new type of target, and its agonists have been proven to be modulators of monoamine neurotransmitters, which are very effective in the treatment of schizophrenia. The three major symptoms of schizophrenia, namely, positive symptoms, negative symptoms, and cognitive impairment, can be treated without producing adverse reactions such as stiffness. What caught our attention is that preclinical studies have shown that TAAR1 agonists are rational for the treatment of depression and anxiety [18,19,20].

SEP-363856 (SEP-856) is a TAAR1 and 5-HT_1A_ receptor agonist, in phase III clinical trials of schizophrenia, which has shown significant efficacy in multiple clinical studies. For example, treatment with SEP-363856 was associated with continued improvement from open-label baseline in the Positive and Negative Syndrome Scale (PANSS) total (−22.6) and the Brief Negative Symptom Scale (BNSS) total (−11.3) scores. In one 4 week trial involving patients with an acute exacerbation of schizophrenia, SEP-363856, a non-D_2_ receptor-binding antipsychotic drug, resulted in a greater reduction from baseline in the PANSS total score than placebo [21,22]. Moreover, it has little effect on body weight, blood lipids, glycemic index, and prolactin, and the risk of extrapyramidal symptoms is also very small. What caught our attention is that one paper showed that SEP-856 could have a significant antidepressant effect in the mouse forced swimming test (FST) [23]. However, no dose-dependent effect was observed. Moreover, there is only that one literature report on the study of SEP-856 on depression; thus, further research on its antidepressant activity is needed.

In summary, both SNRIs and TAAR1 agonists could play a certain role in depression. Therefore, the two types of drugs may have a complex neuropharmacological interaction. The combination of duloxetine and SEP-856 may exert a powerful antidepressant effect. In this innovative thesis, we further discuss the efficacy of duloxetine and SEP-856 in depressive-like behavior in mice, especially their likely interaction with each other, in order to provide a new solution for the treatment of MDD.

## 2. Results

### 2.1. The Effects of Duloxetine and SEP-856 Single or Combination Administration in the FST

It was initially established that independent doses of duloxetine and SEP-856 caused antidepressant-like effects in the FST in male ICR mice. Antidepressant agents enhance escape behavior by decreasing the time of immobility, while the enhancement of immobility time is considered as a depressive-like state. The published dosage range for duloxetine in mouse scientific research applications in FST is 5–42.8 mg/kg p.o. [9,10,24]. The use of 10 mg/kg p.o. was initially tested, and it was determined to significantly reduce the immobility time by 86% (post hoc *p* < 0.0001; Figure 1A). Then, it was proceeded to test smaller doses of 3, 5, and 7.5 mg/kg to determine the dose dependence of antidepressant efficacy (Figure 1A). The ED_50_ of duloxetine was 6.5 mg/kg. In addition, following the published doses of SEP-856 in rodent models of depression, we performed five dose-response experiments to determine the effective dose. [23]. Compared with vehicle, a single oral administration of SEP-856 at 0.3 and 1 mg/kg significantly reduced immobility time (post hoc *p* < 0.01 for the two comparisons; Figure 1B), and the high dose of 10 mg/kg also showed a partial reduction in immobile time (post hoc *p* < 0.05; Figure 1B), indicating that SEP-856 is also likely to exhibit obviously antidepressant effects. However, no significant dose-dependent effect was observed.

Moreover, in order to investigate the effect of SEP-856 on duloxetine, a dose of 0.1 mg/kg without the antidepressant effect of SEP-856 was combined with duloxetine (3, 5, 7.5, and 10 mg/kg, p.o.). Surprisingly, the combination showed more pronounced dose-dependent antidepressant effects (F (1,50) = 275.4, *p* < 0.0001, two-way ANOVA; Figure 1C), and its ED_50_ was 3.8 mg/kg, a decrease by 2.7 compared with duloxetine alone, showing that duloxetine and SEP-856 played a synergistic role in antidepression.

### 2.2. The Effects of Duloxetine and SEP-856 Single or Combination Administration in the TST

The mouse TST, another behavioral test that is vulnerable to all key classes of marketed antidepressants, was used in our experiments. In this test, the mice developed a tendency toward avoidance-led behavior interspersed with temporaryimmobility. Two-way ANOVA showed that a single dose of duloxetine (3, 5, 7.5, and 10 mg/kg) resulted in a nearly dose-dependent significant decrease in immobility time (F (1,35) = 66.53, *p* < 0.0001; Figure 2A), and Bonferroni’s test confirmed that only higher doses (5, 7.5, and 10 mg/kg) significantly reduced immobility time compared to vehicle. In addition, the actual effect of SEP-856 (0.1, 0.3, 1, 3, and 10 mg/kg) was also analyzed in TST (F (1,44) = 16.14, *p* = 0.0002, two-way ANOVA; Figure 2B). Only 0.3 mg/kg administration significantly decreased immobility time (post hoc *p* = 0.0069; Figure 2B).

Furthermore, to confirm whether cotreatment of duloxetine and SEP-856 could provide a synergistic effect, we operated a similar experiment to the FST. Similarly, compared to the single drug of duloxetine, combination therapy also showed a significant dose-dependent antidepressant effect (F (1,44) = 99.05, *p* < 0.0001, two-way ANOVA; Figure 2C), and the ED_50_ reduced from 12.6 to 8.4 mg/kg, reiterating that duloxetine worked well in combination with SEP-856.

### 2.3. The Effects of Duloxetine and SEP-856 Combination Administration on the Locomotor Activity in Mice

In order to detect the harm of the tested chemical substances to the basic motor activity, a locomotor activity test was carried out. A shift in assumptions about the basic motor activity may compromise the presentation of the results obtained in the FST and TST. In addition, the potential calming or hyperactive effects of chemicals can be clearly detected in motor activity tests, which can also indicate a likely suboptimal antidepressant effect. Repeated-measures ANOVA showed that the composition of duloxetine (3, 5, 7.5, and 10 mg/kg) and SEP-856 (0.1 mg/kg) did not alter the motor activity in ICR mice (F (4,35) = 0.515, *p* = 0.7251; Figure 3).

### 2.4. The Effects of Duloxetine and SEP-856 Single or Combination Administration on the SPT in the CUMS Model of Depression

In the sucrose preference test, we further investigated the behavioral effects of duloxetine (15 mg/kg) and SEP-856 (15 mg/kg) administered for 21 days. The sucrose preference tests were performed in the CUMS model of depression. The sucrose preference was expressed as a percentage of sugar water consumption relative to total water consumption. Decreased consumption of sugar water suggested a depressive response in rodents. Furthermore, the dose selection was based on preliminary experiments and previous reports [11,25]. Two-way ANOVA for the sucrose preference (%) revealed a significant effect for stress (F (1,39) = 22.27, *p* < 0.0001) and drug treatment (F (3,39) = 3.384, *p* = 0.0275), as well as a significant effect for the stress × drug treatment interaction (F (3,39) = 5.471, *p* = 0.0031) (Figure 4). Bonferroni’s multiple comparisons test revealed that CUMS induced a significant reduction in sucrose preference compared to the NS vehicle group, (*p* < 0.0001; Figure 4), indicating CUMS induced the depressive state. In the stressed mice, Bonferroni’s post hoc test showed an interaction between CUMS and duloxetine or SEP-856 alone on the sucrose preference, indicating the reversal of the CUMS effect by either or both (*p* < 0.05; Figure 4). Additionally, the statistical analysis of the reversal of CUMS effect in mice co-administered with duloxetine and SEP-856 suggested a decrease in the CUMS-induced anhedonia-like state in these animals (*p* = 0.0019; Figure 4), and the preference rate for the combination increased from 76% alone to 80%, although the increase was not very high.

## 3. Discussion

Duloxetine has excellent antidepression effects in patients with depression, but it may also cause side-effects. Beyond that, ineffectiveness in tolerant patients will limit its application. The development of targeted therapy drugs is an effective way to find new antidepressant drugs. As everyone knows, another strategy may be to use two or more optional drugs with different antidepression efficacy targets at the same time, i.e., synergistic therapy. Our scientific findings suggested that a plausible and reliable way to reduce the use of duloxetine in therapy might be to synergize it with the TAAR1 receptor agonist SEP-856, which also had antidepressant-like effects on its own.

Several classes of chemicals that modulate monoaminergic signaling according to the TAAR1 receptor have demonstrated antidepressant activity in preclinical experiments and in models of depression [19,23]. Here, we investigated the likely interaction between duloxetine and the TAAR1 receptor agonist SEP-856. To better analyze the synergistic potential antidepressant activity of these two chemicals, FST and TST were first implemented, both of which are the most effective dedicated tools for assessing the potential antidepressant activity of chemicals [26]. In this analysis, we confirmed the hypothesis that the combination of duloxetine and SEP-856exerts antidepressant-like activity in mice.

To investigate whether co-administration of duloxetine and SEP-856 affects basal motor activity, a motor activity test was performed, as both too little and too much exercise may profoundly affect the interpretation of results for the FST and TST. Here, in motor activity testing, all chemicals were tested starting at the same time with the same dose schedule as FST or TST. The obtained statistics showed that there was no significant change in the activity test of the mice after the coadministration of the two chemicals. This result clearly suggested that the antidepressant-like effects noted in FST and TST were not compromised by motor activity.

To further investigate the efficacy of duloxetine in combination with SEP-856, the SPT was followed, using the CUMS model, one of the most widely recognized animal models of depression based on chronic stress, in the form of repeated exposure to stressors presented in random order, which provides unpredictability in the type and duration of stressors [27]. Multiple behavioral hazards caused by CUMS in mice have long been described, similarly to some symptoms of depression, such as decreased fondness for sugar water, which is indicative of a lack of happiness [28]. Thus, SPT was selected as a stable method to accurately measure CUMS-induced individual behaviors that are sensitive to chemicals responsible for antidepressant-like effects. This trial showed that duloxetine and SEP-856, independently and synergistically, reversed the CUMS-induced shortcomings of the sugar preference test in mice, indicating antidepressant-like efficacy. However, the effect of the combination treatment was only slightly higher than that of the single treatment, and the effect of the drug also depended on the severity of depression in the mice.

Duloxetine may play an antidepressant role by increasing 5-HT and NE in the synaptic cleft. The antidepressant effects detected in FST and TST acute tests may account for the rapid and significant increase in depression-related neurotransmitters in the brain in rodents. Chronic duloxetine treatment has a long-term regulatory effect on 5-HT and NE pathways, with moderate effects on the release of 5-HT and NE in the hippocampus, which may be responsible for the increased sugar water preference rate [6]. Similarly, the antidepressant effects of SEP-856 may play a similar role, although similar reports are lacking. In addition, although the effect of the combination in the CUMS model was not as significant in the FST and TST, there was a slight increase. The pharmacological effects of drug treatment in CUMS were not strong, which might be the reason that the CUMS model was not built for a long enough time and not put the mice in a severe depression state, such that the therapeutic effects could not be compared. In addition, the reason may also be that the drug treatment time was not long enough or the dose was not high enough to fully reflect the efficacy of the drug. However, it was also possible that the effect of the drug itself in improving the anhedonia-like state in SPT was not very strong. Thus, our first SPT of SEP-856 in the CUMS model just could provide a reference for further research.

The mechanism via which the TAAR1 receptor agonist enhances the antidepressant effects of duloxetine is unclear, but some common mechanisms for the antidepressant effects of the two chemicals can be noted. Some data suggested that, although duloxetine is a SNRI, a TAAR1 receptor agonist appears to play a important role in its antidepressant activity. For example, SEP-856 induced a significant inhibitory response in serotonergic neurons in the dorsal raphe nucleus (DRN) via the 5-HT_1A_ receptor both in vitro and in vivo. The inhibitory response evoked in a subset of ventral tegmental area (VTA) neurons was at least partially based on activating the TAAR1 receptor, suggesting that SEP-856 may regulate serotonergic and dopaminergic neuron firing through distinct mechanisms [29,30]. In addition, modulation of TAAR1 activity by the selective agonist RO5166017 increased the potency of 5-HT_1A_ partial agonists and altered the desensitization rate of 5-HT_1A_ in the DRN [18]. Therefore, co-treatment with TAAR1 agonists may enhance the therapeutic effect of classical antidepressants, further supporting the use of compounds with dual 5-HT_1A_/TAAR1 activity in mood therapy.

Regardless of the mechanism, however, the efficacy of the combination medication is important, and it was found not to affect motor activity These effects strongly suggest that the combination of SNRIs and TAAR1 receptor agonists could be a more effective and safer alternative than duloxetine alone.

Taken together, the TAAR1 receptor agonist SEP-856 may be a good potentiator of the antidepressant effect of duloxetine, which was demonstrated for the first time in our laboratory. This effect may be related to the lower risk of adverse effects with high-dose duloxetine monotherapy. In addition, the antidepressant properties of SEP-856 were also demonstrated for the first time in TST and SPT.

## 4. Materials and Methods

### 4.1. Animals

The experiments were carried out on male ICR mice (Pizhou Dongfang Rabbit Breeding Co., Ltd., Pizhou, China) weighing 20–30 g, 8–12 weeks old, and adapted in a feeding box for 1 week before behavior testing. The animals were group-housed under a 12 h light/dark cycle with free access to food and tap water, maintained at room temperature (21 ± 2 °C) and ambient humidity (50 ± 10% RH) according to An et al. [31]. All individual behavioral tests were performed between 10:00 a.m. and 5:00 p.m., and each tested group consisted of eight mice.

All experimental procedures were carried out in accordance with the basic guidelines of the Small Animal Care Research Unit, Institute of Environmental Health, National University of England. Every effort was made to reduce the total number of small animals applied and to prevent and minimize pain in small animals.

### 4.2. Drugs

Duloxetine and SEP-856 were synthesized by our laboratory. Duloxetine hydrochloride solution and SEP-856 tartaric acid solution were diluted with pure water. Control groups received pure water. All drugs and vehicles were administered orally (p.o.) in a stable volume of 10 mL/kg 60 min prior to behavior testing. For the synergistic treatment of duloxetine and SEP-856, each drug was administered independently. The dosage of drugs was selected on the basis of previous discussions and our pilot trials [4,9,23].

### 4.3. Experimental Procedure

#### 4.3.1. Forced Swimming Test (FST)

When mice (or rats) are put into a limited space and made to swim, they swim desperately to escape at the beginning, before soon becoming floating and immobile, indicating the animals giving up hope of escape and exhibiting behavioral despair. The vast majority of antidepressants or methods, such as tricyclic antidepressants, monoamine oxidase (MAO) inhibitors, most atypical antidepressants, electroconvulsive therapy, and REM (rapid sleep deprivation), are effective against immobility in the context of subacute treatment (2–3-fold within 24 h) and enable the animal to swim. This model is mainly concerned with DA and NA functions, because excitation of DA receptors or α1 adrenergic receptors can reverse the stationary state, while a reduction in DA and NA system functions can enhance the stationary state. In addition, FST is also sensitive to serotonergic drugs [26].

Experiments were carried out in mice according to the procedure detailed in Khakpai et al. [32]. The mice were individually placed in a glass cylinder (25 cm in height, 10 cm in diameter) filled with 15 cm of water and maintained at 23 °C. The animals were left in the water for 6 min. After the first 2 min, the total immobile time was accurately measured during the 4 min test period. When the mouse remained passively floating, it was judged to be motionless.

#### 4.3.2. Tail Suspension Test (TST)

Mice in the suspended tail state soon show desperate behavior, whereby they no longer struggle, but show a unique state of quiet immobility. Antidepressants and central stimulants can significantly shorten the duration of immobility. In order to avoid the interference of central stimulants, it is necessary to use the open-field test to simultaneously measure the autonomous activity of mice, so as to improve the selectivity and reliability of this method for screening antidepressants. The vast majority of antidepressants both reduced the duration of immobility and reduced or did not affect voluntary activity in mice.

The TST was performed in accordance with a previously described protocol [2]. The animal’s tail was immobilized on an iron frame with tape (placed about 1 cm away from the small tail). The duration of immobility was assessed during the last 4 min period similarly to the FST. The mice were in passive relaxation; the body was not exercising, and it was regarded as not moving.

#### 4.3.3. Open-Field Test (OFT)

In order to rule out the possibility that the alteration in the immobility time in the FST and TST was due to interference of the locomotor activity, the spontaneous locomotor activity of each mouse was observed in an open field using video acquisition system software (ZS-ZFT, Huaibei Zhenghua Bio-Apparatus Co., Ltd., Huaibei, China). The device consisted of a gray-black 60 cm × 60 cm × 30 cm container divided into four identical 2 × 2 squares (30 cm × 30 cm). The instrumentation was stored in a dark and acoustically depleted inspection room. Each mouse was located gently on the apparatus, and the distance was recorded in a 6 min interval for a total of 30 min [2]. The apparatus was cleaned with pure water after each experiment.

#### 4.3.4. Chronic Unpredictable Mild Stress (CUMS) Procedure

The CUMS procedural flow was conducted as previously described [12,33], making the required changes. Briefly, our study consisted of 10 different stressors, namely, tail pinch restraint (20 min), forced swimming (25 °C, 20–40 min), white noise (2–12 h), space reduction (2 h), restraint stress (2–4 h), tilted cage (45°, 2–12 h), no bedding (3–12 h), wet bedding (2–12 h), social isolation (4–12 h), and inversion of light/dark cycle for 24 h. Depending on the delay time, two or three stressors were randomly selected and applied daily. The amount of time a stressor was given was changed to maintain the standard of unpredictability, as well as to prevent the same stressor from being picked over and over again. A minimum 2 h rest day was applied between stressors. The stressors were applied for 21 days, and drugs were administered daily.

#### 4.3.5. Sucrose Preference Test (SPT)

After experiencing long-term stress such as CUMS, mice develop depression symptoms such as anhedonia, which can be specifically manifested as a reduced sucrose preference rate in SPT.

SPT was developed as described in previous scientific studies [34,35]. On the first day, the sucrose preference test mice were allowed to consume a 2.5% sucrose solution for 2 h to reduce the risk of new fears. The next day, the SPT officially started. The mice were free to choose between the 1% sucrose solution bottle and the drinking water bottle for 24 h. After 12 h, the bottles were reversed. Mice were fasted from food and water before testing. According to the weight of each bottle before and after the test, the consumption of sucrose and drinking water could be accurately measured. The preference for sucrose water was measured as the percentage of the consumed sucrose solution in the total amount of liquid drunk.

### 4.4. Statistical Analysis

Data were the mean ± SEM, and analysis was performed according to the software GraphPad Prism 8.4.3 (GraphPad Software, San Diego, CA, USA). Statistical analysis significance for FST and TST was assessed on the basis of an analysis of drug efficacy using two-way analysis of variance (ANOVA) with Bonferroni’s multiple comparisons test. Data indicative of locomotor activity were assessed according to repeated-measures ANOVA and Bonferroni’s multiple comparisons test. In the SPT, statistical analysis significance was assessed according to the application of two-way ANOVA and Bonferroni’s multiple comparisons test. Data were considered statistically significant at a confidence limit of *p* < 0.05.

## 5. Conclusions

Our research results showed that duloxetine and SEP-856 had antidepressant effects in mice according to the FST, TST, and SPT. Importantly, the combination of the two drugs could significantly improve the depression-like behavior of mice more than duloxetine alone, and it might provide a new treatment strategy for TRD patients. Moreover, the combination did not affect motor activity. However, the specific mechanism of action needs to be further explored.

## Figures and Tables

**Figure 1 molecules-27-02755-f001:**
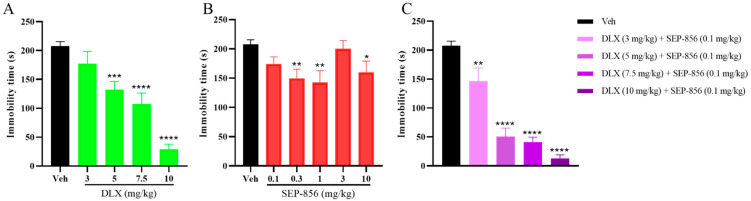
Effects of duloxetine, SEP-856, and their co-administration on the FST in ICR mice. (**A**) Effects of duloxetine (3, 5, 7.5, and 10 mg/kg, p.o.) on behavioral despair in the FST 60 min after administration. (**B**) Effects of SEP-856 (0.1, 0.3, 1, 3, and 10 mg/kg, p.o.) on behavioral despair in the FST 60 min after administration. (**C**) Effects of co-administration of duloxetine (3, 5, 7.5, and 10 mg/kg, p.o.) and SEP-856 (0.1 mg/kg, p.o.) on behavioral despair in the FST 60 after administration. Values are expressed as the mean ± SEM. from 4–8 mice and were analyzed using two-way ANOVA followed by Bonferroni’s post hoc test. * *p* < 0.05, ** *p* < 0.01, *** *p* < 0.001, and **** *p* < 0.0001 vs. vehicle. DLX, duloxetine.

**Figure 2 molecules-27-02755-f002:**
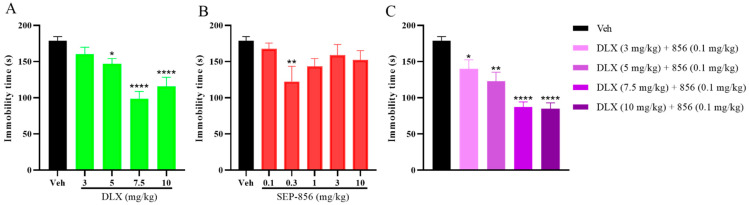
Effects of duloxetine, SEP-856, and their co-administration on the TST in ICR mice. (**A**) Effects of duloxetine (3, 5, 7.5, and 10 mg/kg, p.o.) on behavioral despair in the TST 60 min after administration. (**B**) Effects of SEP-856 (0.1, 0.3, 1, 3, and 10 mg/kg, p.o.) on behavioral despair in the TST 60 min after administration. (**C**) Effects of co-administration of duloxetine (3, 5, 7.5, and 10 mg/kg, p.o.) and SEP-856 (0.1 mg/kg, p.o.) on behavioral despair in the TST 60 min after administration. Values are expressed as the mean ± SEM. from 4–8 mice and were analyzed using two-way ANOVA followed by Bonferroni’s post-hoc test. * *p* < 0.05, ** *p* < 0.01, and **** *p* < 0.0001 vs. vehicle. DLX, duloxetine.

**Figure 3 molecules-27-02755-f003:**
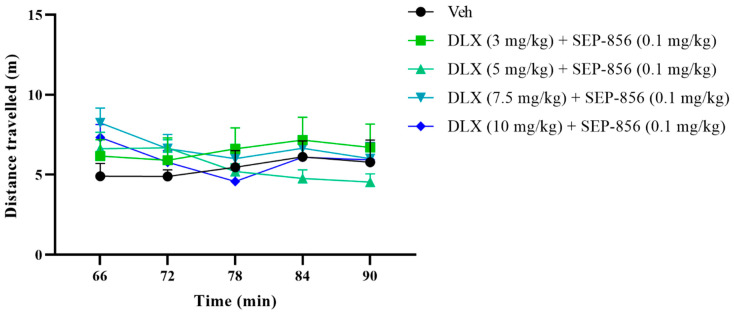
Effects of administration of duloxetine (3, 5, 7.5, and 10 mg/kg) and SEP-856 (0.1 mg/kg) on the locomotor activity of ICR mice. The test was performed 60 min after administration. Values are expressed as the mean ± SEM. from 8 mice and were analyzed using repeated-measures ANOVA followed by Bonferroni’s multiple comparisons test. DLX, duloxetine.

**Figure 4 molecules-27-02755-f004:**
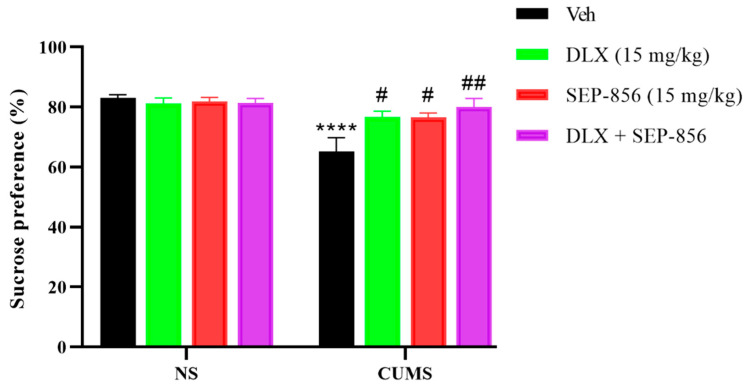
Effects of duloxetine (15 mg/kg), SEP-856 (15 mg/kg), and their co-administration on the sucrose preference in the CUMS model of depression in ICR mice. Values are expressed as the means ± SEM from 5–7 mice and were analyzed by two-way ANOVA followed by Bonferroni’s multiple comparisons test. **** *p* < 0.0001 vs. NS-Veh; # *p* < 0.05, ## *p* < 0.01 vs. CUMS-Veh. NS, non-stressed; CUMS, chronic unpredictable mild stress. DLX, duloxetine.

## Data Availability

Not applicable.

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
