# Peer review of "The Potential Antidepressant Action of Duloxetine Co-Administered with the TAAR1 Receptor Agonist SEP-363856 in Mice"

_molecules, 2022, doi:10.3390/molecules27092755_

Round 1
Reviewer 1 Report
The manuscript describes antidepressant activity of a combined treatment of duloxetine with the TAAR1 receptor agonist SEP-363856 in mice.
The work is supposed to be interesting and the results are interesting, however, the work has serious flaws that must be rectified before further consideration of the publication of this manuscript.
- Has the research been conducted on blinded fashion? This is very important in behavioral research that is very sensitive to the bias of different nature.
- Figure 2 - two-way ANOVA should be used.
- The language requires thorough improvement, the work is written in a language that is difficult to understand. Some examples:
- At present, the pathogenesis of depression is gradually increasing (...)
- (...) a TAAR1 protein kinase inhibitor appears to play a major role in its antidepressant activity regime.
- RO5166017 increased the legal 227 potency of a subset of 5-HT1A agonist
- Every effort was made to reduce the total number of small animals applied to prevent and minimize pain in small animals.
- Leave the animals 267 in the water for 6 minutes.
- The TST was performed in accordance with the previously described agreement [2].
- (...) the soft sugar preference test mice were allowed to consume a 2.5% soft sugar 301 saturated solution for 2 hours to reduce the risk of new fears. (by the way, firstly, the saturated solution has nothing to do with it, secondly, the saturated sucrose solution at 20 oC has a concentration of about 67%)
Reviewer 2 Report
The authors show that oral administration of the SNRI duloxetine and of the TAAR1 Receptor Agonist SEP-363856 exert antidepressant effects and that combined administration of these two substances have synergistic effects, at least in the experiments implying acute administration. Although the subject of study is quite interesting, several methodological issues and interpretation aspects need to be addressed.
Major points:
The most negative part of the study is that it does not seem very well designed as different doses of duloxetine and SEP are used between the FST and TST of depression (both tests performed after acute administration of drugs) and complete new doses are used then for the chronic effects in the CUMS and sucrose preference test. Similarly, different behavioral tests are used to assess the acute versus chronic effects, which makes a comparison of these effects impossible. The authors should provide a rationale for using different doses and for the use of different tests to assess the chronic versus acute effects on depression. If additional results are available (for example FST and TST after chronic administration or different doses - lower doses in the TST), consider including them.
The authors should provide an explanation why much higher doses of duloxetine ( approx. 3 times higher) are needed to see acute antidepressant effects in the TST compared to the FST, whereas for SEP-856 the same doses have quite similar effects. Did the authors also assess the effect of 3, 5, 7.5 mg/kg DLX in the TST or how do they reached to the conclusion that higher doses are needed? This aspect makes the design of the chronic effects (with different doses and different tests) even more problematic.
The effects on locomotion were tested until 30 min after injection (Figure 3), whereas TST and FST were performed 60 min after injection. Although it is probably not expected that DLX-SEP affect locomotor activity after 60 min or at later time points, this can not be excluded and should be shortly discussed. On lines 200-2001 the authors mention that “in motor activity testing, all chemicals were tested on the same duration and usage schedule as FST or TST” which is not the case based on Figure 3.
The open field test used to assess locomotor activity also gives information about anxiety. Please show the time spent in the center of the filed as an indicator of anxiety. DLX have been described anxiolytic-like effects that should also be discussed in the manuscript.
The chronic effects of DLX+SEP on sucrose preference test are not quite fitting to the rest of the data. First, why did the authors use such a high dose of SEP compared to the other experiments and what was the rationale for using the doses of DLX and SEP? And here it can not be seen that a combination of DLS with SEP has better effects (like seen in the FST and TST) than the drugs alone. A p value from <0.05 to <0.01 is not an evidence for an improved effect. The rationale for using the sucrose preference test vs. FST/TST is also not given.
The statistics part is not consistent; once Dunnett’s test once Bonferroni test for the multiple comparisons. I suggest the authors to consistently use the Bonferroni test
The manuscript is quite difficult to follow due to partly poor English. Please send the manuscript to an English editing system.
Minor points:
Include Montgomery–Åsberg Depression Rating Scale on first time mention of the MADRS scale on page 2 line 54
Page 2 line 61: “compared with the use of alone”. Fluoxetine is missing
Page 2 line 64-65: this sentence is confusing “the three major symptoms of spermatozoa do not produce adverse reactions such as stiffness”
Page 2 description of SEP-856. Please give more details about the effects of this agonist (what effects does it have exactly?) instead of writing “significant efficacy in multiple clinical studies”
Page 2 lines 86-87: “Antidepressants increase hit-and-run personal behavior, which in turn reduces dead time, which is considered a depression-like condition”. Please include some introductory information for this test as it is difficult to understand to what the authors refer to.
Page 3 line 100: Depression drug-like efficacy theme activity?
Line 167: the sucrose preference test assesses anhedonia and not apathy
Line 187, 192, 217, 220: SEP is an agonist, right? Here you write it is an inhibitor
Reviewer 3 Report
Points that should be addressed in a revision:
- Lines 47-48: Please rephrase. Traditional antidepressants like duloxetine have an effect latency between 1 and 4 weeks. About one-third of patients do not respond to their first antidepressant drug treatment.
- Line 48: The authors might add: drugs with stronger and more rapid antidepressant effects ...
- Lines 70-76: SEP-856 clinical data derives from a schizophrenia study. The authors reported that the PANSS reduction was greater in the SEP-856 group. However, it would be interesting to know if depressive symptoms in the study participants were also reduced to a greater extend. The authors should report this.
- Line 92 and line 98: The FST does not measure anxiety. The authors should rephrase and write: antidepressant-like behavior instead of anxiolytic-like effects.
- Figures 1-2, 4: The authors reported the use of 2way-ANOVA. However, 1way-ANOVA would be the correct statistical test here. Please correct.
- Figure 4: Did the authors test whether all animals detected sucrose sufficiently aka that all mice were statistically different from chance level (50%)?
Reviewer 4 Report
Authors explore the effect of the combination of the antidepressant duloxetine with a Trace amine-related receptor 1 (TAAR1), SEP 856, in three models used for the screening of antidepressant drugs. This design represents a strength of the study because increases the convergence of the effect however the are some issues that require attention before to published.
- The authors mentioned in line 98 the use dependence of anxiolytic efficacy when are describing antidepressant-like activity?? Is confuse, please review.
- Dose selection of duloxetine and SEP 856 was based on previous reports, please incorporate the reference line 168.
- There is no synergism in CUM model. How to explain this? The authors mention that the effect (an increase in sucrose preference) is not very high, demonstrating the advantages of the combination…line 181. How is an advantage the fact of not observing a synergism as occurs in acute stress models? It could be expected that in three models (FST, TST and CUM) the combination was effective. However, the results obtained with CUMS are not an additive effect since results are similar to those induced by each drug alone. These results affect the convergence between models. A discussion on the sensitivity of models will contribute to explaining the results. Furthermore, there are differences in the ED50 of duloxetine in FST and TST, may occur the same in CUM?
- Review the discussion section, lines 222-224, since, from the present results, an argument in favor of a synergic effect of the combination of duloxetine plus SEP 856 can´t be elaborate. The authors have a ceiling effect that needs be discussed.
- According to the literature, FST is also sensitive to serotonergic drugs, and this information needs to be included. Please review Cryan and Lucki's references.
- Do use independent groups of mice in each experiment? How many animals per group? What was the rationale behind repeating the locomotor activity test?
Round 2
Reviewer 1 Report
The authors have improved the manuscript according to my suggestions.
Reviewer 2 Report
The major problem with the study was the use of different doses for the acute effects of duloxetine in the FST and TST, and completely new (much higher) doses for the chronic effects which were tested in the SPT (again another test) and not in the FST and TST, making any comparison impossible. The authors say they repeated the experiments with duloxetine and TST (Figure 2A, 2C) and replaced this Figure with new doses that now match the doses used in Figure 1A and 2C (FST). This newly added Figures do not seem very trustworthy. While the Veh are absolutely the same, the dose of 10 mg/kg duloxetine (Figure 2A) had in the original version no antidepressant effect at all and now it is highly antidepressant (p<0.0001). A similar situation can be also observed in Figure 2C – Vehicles identical and DLX (10 mg/kg) + SEP (0.1 mg/kg) originally at around 120 s immobility time (p<0.05) and now around 80 s immobility time (p<0.0001). Although substances might have slightly different effects when tested in different batches, these results are not trustworthy especially because of the identical Vehicles (these should also slightly vary from batch to batch). The authors argue that the SPT was used for the chronic experiments as the FST and TST can be used for the acute antidepressant effects and SPT after modeling of depression – this is not the case, FST and TST can also be used for chronic effects of drugs and in models where depression was induced. The rationale for using much higher doses is not given and data showing that lower doses (like used for the acute effects) are ineffective are missing. The authors also do not show the anxiety levels in the OFT because no such effects were found and these effects are not supported by the literature. Even so, these should be shown and discussed.
Comments:
Line 15: add „especially at doses of 0.3…”
Line 45: add “mein adverse reactions include negative effects on the cardiovascular system…”. Remove “aspects” in line 46
Line 66: three major symptoms of schizophrenia, namels …. Please include them
Line 73: what is the full name of PANSS and BNSS score?
Line 78-79: you mention “in one paper”, “only that one literature report”… and cite 2 papers
Line 88: “with a view to providing MDD The diagnosis and treatment give new plans”?
Line 105-106 “However, no significant usage-dependent effect 105 was observed” What are the authors referring to?
Major: Line 163: provide here the rationale why SPT was used here (and not FST and TST) and why such higher doses than in the acute tests. FST and TST can be used after both acute and chronic administration. The argument that the doses were based on previous reports and preliminary experiments (lines 168-169) makes no sense here, as different doses were used in Figure 1 and 2, which actually are the preliminary studies for the chronic effects. Please show these preliminary experiments to make your study “sound” – otherwise the rationale is missing and the authors might consider submitting the acute and chronic tests separately.
Line 181: please remove ”which also demonstrated the advantages of the combination” – this is not supported by the results shown in Figure 4
